# Adaptive Complexity: Examining Texas Public Postsecondary Institutions' Provision of Student Basic Needs Programs

Lisa K. Zottarelli [1,*], Xiaohe Xu [2], Jayla M. Hatcher [2], Raji Thiruppathiraj [2], Natasha Ellis [3], Shamatanni Chowdhury [4] and Thankam Sunil [4]

[1] College of Social Work, University of Tennessee, Knoxville, TN 37916, USA
[2] Department of Sociology and Demography, University of Texas at San Antonio, San Antonio, TX 78249, USA; xiaohe.xu@utsa.edu (X.X.); jayla.hatcher@my.utsa.edu (J.M.H.); raji.thiruppathiraj@my.utsa.edu (R.T.)
[3] Department of Sociology, University of Tennessee, Knoxville, TN 37916, USA; bht288@vols.utk.edu
[4] Department of Public Health, University of Tennessee, Knoxville, TN 37916, USA; schowd15@vols.utk.edu (S.C.); tsunil@utk.edu (T.S.)
* Correspondence: lzottare@utk.edu

**Abstract:** This study reveals that the likelihood and diversity of postsecondary institutions providing basic needs programs are significantly influenced by institutional factors such as the institutions' organization and size. This study also indicates that Hispanic-Serving Institutions tend to provide emergency housing, which highlights a targeted response to specific community needs. In addition, the analysis indicates that the presence of students with financial needs is linked to the availability of food pantry services, suggesting a strategic approach to address student welfare. The findings from this study provide critical insights into how institutional characteristics influence the provision and variety of basic needs services. These conclusions not only underscore the pivotal role of such services in supporting the overall well-being and academic success of students but also indicate institutional factors that support the formal implementation of a variety of basic needs programs to meet diverse student needs.

**Keywords:** basic needs insecurities; basic needs initiatives; student support programs; organizational response; organizational structure





## 1. Introduction

Colleges and universities must respond to a more diverse and expanded postsecondary student population; a population with needs that differ from "traditional" college students, who are conceptualized as economically dependent upon parents who have the resources to support them. Given the costs of attending college, even students categorized as traditional in terms of age at their first time in college and dependency status may find themselves experiencing financial difficulties in covering expenses. Insufficient financial resources can be extreme and may result in poor academic performance and/or failure to continue toward degree completion [1].

A lack of financial resources resulting in unmet basic needs can disrupt enrollment, reduce retention, and/or delay graduation [2–5]. Some colleges and universities have responded to students' unmet basic needs by developing and formalizing basic needs programs [6–8]. To date, research has focused primarily on the prevalence of students with unmet needs, with an emerging body of literature on institutional responses. There is a lack of research that examines institutional factors that may influence the implementation of student unmet basic needs programs. To that end, we seek to describe institutional factors associated with the presence of formal programs designed to address unmet student basic needs across institutions.

## 1.1. Unmet Basic Needs among College Students in the United States

Access to postsecondary education has expanded to include more diverse student populations than in previous generations, and there is increasing attention given to student success as evidenced by student retention and degree completion rates [9]. While completion of postsecondary education is attributable to several factors, basic needs insecurities have been identified as barriers to student academic success [10,11]. As Maslow established, basic physiological needs, such as food, shelter, and safety, must be met before higher-order needs can be pursued and secured [12]. Known predictors of student success, such as academic preparation [13,14], as well as non-cognitive factors, such as a sense of belonging [15,16], goal engagement, and self-efficacy [14], are established on the presumption of needs for food, shelter, clothing, etc., (i.e., basic needs) having been met.

Food and housing insecurities among college students are found across U.S. postsecondary institutions [17–19]. Students attending postsecondary institutions are more likely than the general population to be food insecure [20], and many students who are classified as independent live below the poverty guidelines [21]. A lack of resources to meet basic needs places students at greater risk of experiencing difficulties in attending classes and performing to their full academic ability [2–5]. Students who experience food insecurity are more likely to have lower grade point averages [4,22]. In addition, food insecurity tends to increase poor mental health outcomes among college students, which also hinders students' academic performance [2,4] and may increase the odds of stopping out [23]. By addressing students' basic needs insecurities, postsecondary institutions are seeking to improve student retention and degree completion [10].

Research on basic needs among students in higher education initially focused on food insecurity and expanded to include housing insecurity [24]. More recently, research efforts have further expanded to examine a variety of basic needs insecurities that make going to college difficult. These include mental health problems [19,23–26], lack of access to transportation [19,24], and lack of access to childcare [19]. The interconnected nature of basic needs insecurities among students is becoming more apparent, as is the need for more complex and comprehensive institutional responses. Food and housing insecurities are often coupled with other unmet basic needs, and the multifaceted nature of the unmet needs creates challenges for institutions to develop comprehensive student service programs [27].

## 1.2. Institutional Efforts to Address Basic Needs Insecurities

Over the past decade or so, postsecondary institutional efforts to address basic needs insecurities among college students have gained ground. Colleges and universities have been entrepreneurial in developing context-specific programs for their students. Examples, approaches, and how-to information are readily available to support the establishment of food pantries on college campuses [7,8,28]. Information sharing in the forms of campus visits and presentations at national conferences and meetings has become increasingly common.

A few large postsecondary systems and states have started to require institutional efforts to address basic needs insecurities within their college and university campuses. The University of California system has created a healthy campus network initiative to address the multidimensional issues of food insecurity, emotional well-being, and other basic needs [29]. The state of California allocated USD 100 million to support the establishment of basic needs centers at community colleges throughout the state [6]. The state of New York mandates food pantries or stigma-free food access on the State University of New York and City University of New York campuses [30,31].

The categories of services being offered are extensive. A case study on the Florida College System found numerous basic needs services, including such programs as food and housing, professional clothing, eyeglasses, financial and legal counseling, bus passes, childcare, and disaster relief [32]. Within the California State system, a variety of programs have been established to address food and housing insecurities through food pantries, hot meal programs, emergency housing, rental assistance, and referrals to off-campus social services [11]. In Texas, Hispanic-Serving Institutions offer programs to address unmet

needs for food, housing, transportation, childcare, clothing, emergency financial assistance, employment services, social services, and physical and mental health services, with most institutions offering five or more categories of programs [33,34].

*1.3. Professionalization of Basic Needs Services at Postsecondary Institutions*

In response to mandates from states and postsecondary systems, and to coordinate the number and types of services, basic needs programs at some institutions are becoming formalized, with dedicated staff and other institutional supports. Institutions that provide a wider array of services tend to have multilevel administrative support at the presidential and vice-presidential levels, along with staff and infrastructure investment [11]. Student affairs professionals have begun and will continue to develop and coordinate interconnected and comprehensive basic needs programs [27]. Directors are being hired to oversee and coordinate basic needs services at institutions across the country [35]. For example, all California community colleges have a basic needs coordinator and a basic needs center [36]. In California and elsewhere, student affairs professionals, social workers, and other social service professionals are employed to work with college students and to connect students to a variety of basic needs services [10,27].

In the absence of organized efforts at the state or system level, many colleges and universities have become entrepreneurial and intentional in formalizing approaches to addressing basic needs insecurities among their students. This includes entering into agreements with local food banks and other community-based agencies to provide ongoing support for these efforts and centralizing services to better address multiple student needs.

*1.4. What Can Explain Institutional Responses to Student Basic Needs?*

Effectively addressing student basic needs requires comprehensive services [17,27]. Yet, student needs alone are unlikely to explain the presence of specific types of services. For example, though housing insecurity is frequently experienced by college students [19,24,37], few institutions provide emergency housing or alternative housing programs [34,38]. As such, it is essential to understand how institutional characteristics may explain the presence (or absence) of basic needs programs.

Guided by Rouse's use of Snowden and Boone's and Poli's frameworks, we examine the implications of colleges and universities as complex adaptive systems consisting of both complicated system factors and complex system factors [39–41]. Elements of complicated systems include expertise, clearly defined roles, and operational conditions, which can be "decomposed" into organizational charts and procedures. On the other hand, complex systems emerge from practice, creativity, and the ability to change in response to external societal factors beyond the control of the institution. This juxtaposition of the presence of both complicated and complex system factors within an institution creates "complex adaptative systems" that are diverse and engage in problem-solving in ways that align with organizational structure and culture to produce emergent responses to identified challenges [39,42].

Institutions exist within a broader societal and governmental ecosystem. The structure of the institution, consisting of the ways in which each institution is organized across academic and non-academic units, shapes policy and practices. This architecture of the system, as Rouse conceptualizes, and the elements comprising that architecture act to enable or inhibit other elements within the system [39]. Economic elements are integrated into the architecture. We argue that complex adaptive systems, comprising complicated system factors and complex systems factors, influence institutional responses in the form of policies and practices. Further, complicated system factors and complex system factors function differently within a single institution. Organizationally, given the nature of complicated system factors, it is more likely that those factors can produce formal organizational responses to situations such as unmet basic student needs.

Using complex adaptive systems to inform and frame this study, we consider factors within a postgraduate institution that would reflect both complicated system characteristics

and complex system characteristics. The complicated system characteristics or factors we consider include institutional size and setting, the number of physical locations offering all or part of a degree or certificate program, and the level at which the institution is authorized to offer degrees. These characteristics reflect the institutional capacity to manage a range of services, which can be decomposed into organizational charts and procedures and require organizational expertise. The second set of characteristics or factors we examine reflects the institution as a complex system featuring societal issues within the institutional context. These factors include the economic needs of students, the socioeconomic diversity of students, and historical underrepresentation of students.

Although both complicated and complex characteristics can contribute to problem-solving and addressing needs for change, colleges and universities can misconstrue complicated characteristics for complex ones [39]. Utilizing complicated systems that recognize expertise and integration into institutional response in terms of the assignment of responsibilities and documented procedures is more likely to result in formal programs and services. Therefore, we hypothesize that complicated system characteristics or factors are more likely to be associated with the presence of basic needs programs than complex system characteristics or factors within Texas public postsecondary institutions.

To test this hypothesis, we focus on the seven areas of basic needs programs and explore the multifaceted linkages between institutional characteristics and the presence of these basic needs programs at public postsecondary institutions in the state of Texas. Texas is a large and diverse state with more than 100 independently accredited public postsecondary institutions. The state does not require that public colleges and universities provide basic needs programs for students. Since public postsecondary institutions in Texas are governed by the same higher education coordinating board, the same state legal requirements, and the same uniform funding models, the presence or absence of basic needs programs can be reasonably attributable to variations in institutional characteristics.

## 2. Materials and Methods

In this study, we examine postsecondary institutional responses to students' unmet basic needs using a quantitative content analysis of institutional websites and secondary data from three administrative sources. We conceptualize basic needs broadly, consistent with research by Nix et al. [32] and our previous research [33] on the array of available services. We identified and included seven areas of basic needs programs in this study. Data were collected from fall 2022 to spring 2023. Due to the aggregate and public-use nature of the data, Institutional Review Board approval was not required.

A quantitative content analysis of institutional websites was conducted because of the important role that institutional websites have in representing the institution and serving as a communication channel to students. The visual and textual information provided on websites is intentional and communicates institutional priorities [43]. The official websites are curated repositories of information that students can use to find information about a variety of policies, programs, and services. Postsecondary institutions control their institutional websites and limit the information posted. Therefore, we would expect only those basic needs programs that are officially endorsed by the institution to be included on the website.

### 2.1. Population

To identify Texas public 2-year and 4-year institutions, we made use of the 2022 institutional database maintained by the Southern Association of Colleges and Universities Commission on Colleges (SACSCOC) [44]. After identifying all public postsecondary institutions in Texas, we removed graduate-only and health science centers from the database. Thus, the remaining 103 independently accredited public institutions constituted the analytical sample for analysis.

*2.2. Data*

To collect data on basic needs programs at colleges and universities, we conducted a quantitative content analysis of institutional websites that support intentional information seeking with structures such as search tools and directories. Institutional websites typically link students to the learning management system, provide information on services such as tutoring and library resources, and are the repository for documents such as official catalogs and academic calendars. The presence of basic needs programs on an institution's website is an indication of formalized institutional support as well as an attempt to communicate the presence of the services to the campus community.

To ensure the accuracy, completeness, and reliability of the data collected, members of the research team were trained in the website data collection protocol. Data collection involved a series of systematic keyword searches utilizing the website search bar, which was often located on the landing page. The list of basic needs programs included as keywords was based on previous research [33]. Additionally, the keywords of "basic needs", "student needs", and "help" were also used. Starting at the landing page, the keyword was entered. The results of the keyword search were read, and the links on the first page of results were followed to determine if the information provided was an indication of a basic needs program at that institution. Two members of the research team collected data from each institutional website independently, and inter-rater reliability was closely monitored. When discrepancies in data collection were identified, a third member of the research team reviewed the website independently. When discrepancies remained, members of the research team met to review adherence to the data collection protocol and to come to a consensus on the data in question. This process continued until all inconsistencies were resolved. As a result of the quantitative content analysis, the data were organized into the following categories of basic needs programs: (1) food pantry, (2) emergency housing, (3) childcare, (4) financial emergency assistance, (5) mental health counseling, (6) transportation, and (7) other assistance. The category of "other assistance" referred to a variety of basic needs programs offered at only a few institutions. Examples in this category included a help desk to direct students to off-campus services, legal services for students, and clothing assistance.

Data for complicated system characteristics were collected from administrative sources. First, the Carnegie Classifications of Institutions of Higher Education Data File [45] was utilized. In the file, the size and setting classifications recognize academic institutions by (1) the length of study (i.e., time to degree) and academic level (i.e., 2-year and 4-year), (2) undergraduate enrollment size (i.e., very small/small to very large), and (3) residential status (i.e., percentage of full-time degree-seeking undergraduates living in institutionally owned, operated, or affiliated housing). The classification results in 17 categories. The number of students, housing infrastructure, and the level of degree programs are indications of the institutional capacity to manage infrastructure and offer a range of service delivery.

The second complicated system characteristic is the highest degree authorized. These data were collected from the SACSCOC Institutional Database [44]. In Texas, almost all traditionally designated 2-year public institutions offer at least one bachelor's degree in addition to certificates and associate degrees for vocational, technical, and transfer programs. The addition of a bachelor-level program requires a substantive change approval, and one aspect of that process is the review of the institutional capacity to provide services such as academic advising, library, career counseling, and tutoring appropriate to meet the needs of the bachelor's program. Despite being historically and primarily 2-year institutions, these 2-year institutions are classified as bachelors-granting institutions. In a similar fashion, Texas 4-year institutions frequently offer at least one graduate degree in addition to offering programs at the associate and bachelor's levels. These institutions have to go through a substantive change process to demonstrate their capacity to support programs at the highest level offered.

The third complicated system characteristic is the number of off-campus instructional sites. These are locations that are physically removed from the main campus with at least

25% of a degree or certificate program offered. In many cases, off-campus instructional sites offer 50% or more of a degree or certificate program. These sites often lack direct service delivery and rely on online access or expect students to visit the main campus or a campus with more resources as needed. The activities on the site have to be coordinated with the main campus and delivered on the site. The more off-campus instructional sites there are, the more complicated it becomes, and greater volume and structural arrangement is necessary to effectively coordinate activities. Data on the number of off-campus sites came from the SACSCOC Institutional Database [44].

Data for complex system characteristics were collected from administrative sources as well. We utilized the Undergraduate Profile Classification (UPC) from the Carnegie Classifications of Institutions of Higher Education Data File [45]. The UPC has 15 categories comprising 2-year and 4-year institutions, percentages of part-time/full-time student status, inclusive or selective access, and percentage of transfer students. The UPC describes differences in the undergraduate population, and these differences have implications for student needs across a range of services and outcomes within an institution. These are complex system factors because the characteristics of the students reflect conditions external to an institution. While the student population characteristics have implications for the scheduling of classes, time to degree, advising needs, and extracurricular activities, the students are not simply mapped onto an organizational chart or reflected in a process or procedure.

To gauge the overall economic status and financial needs of the enrolled students, the percentage of students receiving Pell Grants was obtained from the Texas Higher Education Coordinating Board [46]. The Hispanic-Serving Institutions (HSIs) designation data were obtained from the United States Department of Education [47]. These institutions are designated by the United States Department of Education based on having at least 25% Hispanic undergraduate full-time-equivalent students. Many postsecondary institutions in Texas are designed as HSIs. While HSIs can vary significantly in size, they often enroll a higher proportion of students who are first-generation college students and from socioeconomically disadvantaged backgrounds.

### 2.3. Measures

The identified basic needs programs (i.e., (1) food pantry, (2) emergency housing, (3) childcare, (4) financial emergency assistance, (5) mental health counseling, (6) transportation, and (7) general student help/assistance) were used as the dependent variables. All of these variables were dummy-coded, with 1 meaning the program was offered and 0 meaning the program was not offered as of the spring semester of 2023. However, due to the lack of variation, mental health counseling programs were excluded from the basic needs-specific multivariate statistical analyses because 100 out of 103 postsecondary institutions offered mental health counseling services. In addition to the dummy-coded individual basic needs program variables, a count variable was created to capture multiple basic needs programs offered.

### 2.4. Complicated System Variables

The size and setting variable was recoded from the original 17 categories into an ordinal variable ranging from 1 for institutions classified as 2-year, small/very small, and non-residential to 9 for institutions classified as 4-year, large, and primarily residential. The highest degree offered was coded into an ordinal variable as follows: 1 = associate degree, 2 = baccalaureate degree, 3 = master's degree, and 4 = doctorate degree. The number of off-campus sites was a metric variable ranging from 0 to 150 locations.

### 2.5. Complex System Variables

The Undergraduate Profile Classification (UPC) was recoded from the 15 original categories into a categorical variable as follows: 1 = 2-year-part time, 2 = 2-year mixed, and 3 = 4-year. This variable was further dummy-coded, with category 3 (i.e., 4-year) serving as

the reference. The Pell Grant variable is the percentage of students receiving Pell Grants at the institution. The indicator of HSIs was dummy-coded into 1 = yes and 0 = otherwise.

### 2.6. Analytical Strategies

In addition to descriptive analysis for all the variables included in this study, binary logistic regression models were used to estimate the independent effects of the characteristics of Texas public postsecondary institutions on the likelihood of offering each basic needs program. In these regression models, the dichotomized HSI variable was used as an independent variable as well as a statistical control. Extra-dispersed Poisson regression models were estimated for the count variable that combined all the basic needs programs. A statistical significance of $p < 0.05$ was employed to highlight the relative importance of the institutional characteristics.

Due to multicollinearity problems, only one institutional characteristic (other than the HSI dummy variable) was estimated for each of the 7 dependent variables (six basic needs programs, excluding the mental health counseling program, and one combined count variable). To ease interpretations, all the estimated regression coefficients were exponentiated. These exponentiated regression coefficients represented the odds ratios in binary logistic regression models and the expected count or number of basic needs programs in Poisson regression models, respectively. All statistical analyses were conducted using Stata 17 [48].

### 3. Results

Table 1 reports descriptive statistics for all the variables included. As shown in the table, the majority (75.7%) of Texas public postsecondary institutions included were HSIs. On average, these HSIs were 4-year, small, and primarily nonresidential colleges or universities that offered a baccalaureate degree. The included colleges and universities on average owned and operated 34 off-campus instructional sites. On average, 34% of the enrolled students received federal Pell Grants.

**Table 1.** Descriptive statistics.

|  | Number | Percent | Mean | Standard Deviation |
|---|---|---|---|---|
| Food Pantry |  |  |  |  |
| No | 20 | 19.4 | - | - |
| Yes | 83 | 80.6 | - | - |
| Emergency Housing |  |  |  |  |
| No | 20 | 19.4 | - | - |
| Yes | 83 | 80.6 | - | - |
| Childcare |  |  |  |  |
| No | 24 | 23.3 | - | - |
| Yes | 79 | 76.7 | - | - |
| Financial Emergency Assistance |  |  |  |  |
| No | 17 | 16.5 | - | - |
| Yes | 86 | 83.5 | - | - |
| Other Assistance |  |  |  |  |
| No | 26 | 25.2 | - | - |
| Yes | 77 | 74.8 | - | - |
| Transportation |  |  |  |  |
| No | 35 | 34.0 | - | - |
| Yes | 68 | 66.0 | - | - |
| Mental Health |  |  |  |  |
| No | 3 | 2.9 |  |  |
| Yes | 100 | 97.1 |  |  |
| All BNI Programs Combined |  |  |  |  |
| 0 | 2 | 1.9 | - | - |
| 1 | 4 | 3.9 | - | - |

**Table 1.** *Cont.*

|  | Number | Percent | Mean | Standard Deviation |
|---|---|---|---|---|
| 2 | 3 | 2.9 | - | - |
| 3 | 5 | 4.9 | - | - |
| 4 | 8 | 7.8 | - | - |
| 5 | 12 | 11.7 | - | - |
| 6 | 24 | 23.3 | - | - |
| 7 | 45 | 43.7 | - | - |
| Hispanic-Serving Institution |  |  |  |  |
| No | 25 | 24.3 | - | - |
| Yes | 78 | 75.7 | - | - |
| Undergraduate Profile Classification |  |  |  |  |
| 2-Year Mixed | 10 | 9.7 | - | - |
| 2-Year Part Time | 52 | 50.5 | - | - |
| 4-Year Institutions | 41 | 39.8 | - | - |
| Institutional Size | - | - | 3.9 | 2.6 |
| Highest Degree Offered | - | - | 2.2 | 1.3 |
| Off-Campus Sites | - | - | 34.0 | 36.9 |
| Receiving Pell Grants (%) | - | - | 34.0% | 11.1% |

Turning to the basic needs programs, the most popular basic needs programs offered by Texas public postsecondary institutions were mental health counseling programs (97.1%), which were followed by financial emergency assistance programs (83.5%), emergency housing programs (80.6%), and food pantry programs (80.6%). While transportation programs were the least common (66%), childcare programs (76.7%) and other assistance (74.8%) were notable as well. Nearly 44% of Texas public postsecondary institutions included in this study offered all seven basic needs programs, whereas slightly more than 23% offered six basic needs programs. In effect, 98% of these colleges and universities offered at least one program.

Table 2 displays the exponentiated regression coefficients of the binary logistic and extra-dispersed Poisson regression models to predict the odds and number of basic needs programs. As can be observed from Panel 1 in Table 2, being either a 2-year college characterized by both part-time and full-time students or a 2-year college primarily comprising part-time students was associated with decreased odds of offering individual basic needs programs, as well as a decrease in the number of basic needs programs. Regression coefficients were statistically significant at the 0.05 or 0.01 levels for food pantry, transportation, and the total number of basic needs programs. USP: two-year mixed was statistically significant for emergency housing, and USP: two-year part-time was statistically significant for childcare and other assistance. Panel 2 reveals more consistent findings. That is, one unit increase in institutional size was significantly associated with increased odds of offering basic needs programs and an increase in the number of basic needs programs (all regression coefficients were statistically significant at a level of at least 0.05). In a similar fashion, the regression coefficients displayed in Panel 3 show that the highest degree offered was positively and significantly associated with the odds and number of basic needs programs (the coefficients were significant at a level a of at least 0.05). On the other hand, as shown in Panel 4, the number of off-campus sites decreased the odds of basic needs program offerings, such as food pantry programs ($p < 0.05$) and transportation programs ($p < 0.05$). The regression coefficients included in Panel 5 reveal positive but very weak associations between receiving Pell Grants and the odds and number of basic needs programs. That is, a one-unit increase in the percentage of students receiving Pell Grants was significantly associated with increased odds of there being food pantry programs ($p < 0.05$). Finally, being a Hispanic-Serving Institution increased the odds of offering emergency housing programs, net of other institutional characteristics. This positive regression effect was statistically significant at either the 0.05 or the 0.01 level (see the third column in Table 2).

**Table 2.** Exponentiated coefficients of binary logistic and extra-dispersed Poisson regression models to predict odds and number of programs among Texas's colleges and universities (n = 103).

| Independent Variable | Food Pantry [a] | | Emergency Housing [a] | | Childcare [a] | | Financial Emergency [a] | | Transporation [a] | | Other Assistance [a] | | Total Number of Programs [b] | |
|---|---|---|---|---|---|---|---|---|---|---|---|---|---|---|
| HSI | 0.821 | | 3.894 | * | 0.918 | | 2.241 | | 1.363 | | 1.056 | | 1.062 | |
| USP: Two-Year Mixed | 0.077 | ** | 0.185 | * | 0.184 | | 0.179 | | 0.171 | * | 0.324 | | 0.733 | * |
| USP: Two-Year Part Time | 0.142 | * | 0.419 | | 0.150 | ** | 0.263 | | 0.193 | ** | 0.261 | * | 0.793 | ** |
| Model $\chi^2$ | 11.47 | ** | 9.75 | * | 11.09 | *** | 6.68 | | 12.60 | ** | 6.69 | | 11.87 | ** |
| HSI | 0.751 | | 4.253 | * | 0.772 | | 2.135 | | 1.261 | | 0.964 | | 1.065 | |
| Size and Setting | 1.443 | ** | 1.435 | ** | 1.617 | ** | 1.383 | * | 1.430 | *** | 1.445 | ** | 1.054 | *** |
| Model $\chi^2$ | 9.52 | ** | 14.93 | *** | 15.95 | ** | 8.30 | * | 14.76 | *** | 11.97 | ** | 16.30 | *** |
| HSI | 0.908 | | 4.572 | ** | 0.949 | | 2.607 | | 1.444 | | 1.099 | | 1.084 | |
| Highest Degree Offered | 1.808 | * | 1.620 | * | 1.686 | * | 1.929 | * | 1.654 | ** | 1.519 | * | 1.089 | ** |
| Model $\chi^2$ | 6.82 | * | 10.12 | ** | 6.47 | * | 8.38 | * | 7.94 | * | 4.59 | | 8.95 | ** |
| HSI | 1.076 | | 3.281 | * | 0.904 | | 1.776 | | 1.527 | | 0.934 | | 1.063 | |
| Number of Off-Campus Sites | 0.987 | * | 1.001 | | 0.993 | | 1.003 | | 0.988 | * | 0.998 | | 0.999 | |
| Model $\chi^2$ | 4.52 | | 5.16 | | 1.30 | | 1.33 | | 4.57 | | 0.10 | | 1.41 | |
| HSI | 0.868 | | 3.856 | * | 0.829 | | 2.087 | | 1.208 | | 0.925 | | 1.050 | |
| Percent of Pell Grant Recipients | 1.096 | ** | 1.042 | | 1.037 | | 1.035 | | 1.035 | | 1.005 | | 1.006 | |
| Model $\chi^2$ | 8.50 | * | 7.92 | * | 6.47 | * | 3.01 | | 3.02 | | 0.08 | | 3.65 | |

* $p < 0.05$. ** $p < 0.01$. *** $p < 0.001$. [a]. Binary logistic regression. [b]. Extra-dispersed Poisson regression.

## 4. Discussion

This study adds to the literature on student basic needs insecurities by examining the institutional characteristics associated with the presence of basic needs programs in the state of Texas. Consistent with previous research [24,32,33], our analysis indicated that Texas public postsecondary institutions were capable of providing a wide array of basic needs services for students. Indeed, as revealed by our results, the majority of colleges and universities included in this study offered between five and seven types of basic needs services for their students. More notably, mental health programs were readily available at all but three institutions as of 2023. Even in a state such as Texas with no state-wide mandates that require basic needs programs, individual colleges and universities have established and budgeted for basic needs programs, suggesting institutional awareness of the widespread challenges resulting from basic needs insecurities among college students. Furthermore, the presence of multiple types of basic needs programs could lead to increased student awareness and access, as accessing one service can serve as a gateway to accessing other basic needs services as well as academic and social support services [17].

Unlike previous research that has used surveys of administrators to identify the presence of specific basic needs programs [24,32–34], we used institutional webpages to identify programs. We chose to use this data collection strategy because it approximates how students may search for information about services. Additionally, the presence of a basic needs program on the institutional webpage is an indication of formal institutional support for that program as well as an indication of program maturity.

Both survey and website content analysis yielded similar results in terms of the variety of basic needs programs. However, the use of website content analysis brings in the lens of students and others in the campus community who may search for this information. It signifies actions taken by an institution to communicate to students who experience basic needs insecurities and make them aware of available programs and services to address their needs. By formally including basic needs programs in their websites, institutions are potentially communicating an assurance of belonging that is important to academic success [15,16].

We incorporated a theoretical perspective to frame and explore how institutional characteristics affect institutional responses. All institutions in higher education exhibit a combination of complicated and complex characteristics, and the combination promotes institutional capacities and responses. We hypothesized that complicated institutional factors would be associated with the presence of basic needs programs, while complex institutional factors may not. Our study findings supported this hypothesis. That is, the organizational capacity measures, such as size and setting and the highest degree offered, are important institutional characteristics that are associated with formal institutional responses to college students' basic needs insecurities.

Complicated characteristics indicate an institutional capacity to organize, distribute, and manage resources throughout the institution [39]. They could also indicate technical approaches to problem-solving that result in formal and structured responses, which in turn may be more likely to appear on the institutional website. Likewise, as complicated characteristics rely on expertise and specialized roles, these can be directed at specific actions, such as basic needs program development and management.

As anticipated, complex institutional characteristics were not consistently and systematically associated with the presence of basic needs programs. Our regression results showed that the percentage of students receiving Pell Grants was significantly associated with food pantry program offerings, whereas HIS status was positively associated with emergency housing programs. Although less systematic, these are important findings. It might be the case that Texas public postsecondary institutions developed and provided food pantry programs as responses to food insecurity experienced by a growing number of college students, whereas HSIs prioritized emergency housing programs to address critical and increasing needs (e.g., homelessness) for minority students in addition to food pantry programs [18].

The complex characteristics within higher education institutions reflect broader societal conditions, and the identification of these societal conditions, such as basic needs insecurities [17–19,23–25], has been the focus of research on college student basic needs. These societal conditions (i.e., Rouse's ecosystem) act as constraints within which the institution operates. It is within the mission, vision, value, and high-level strategic planning where complex characteristics occur within higher education institutions. Therefore, while student needs serve as the context, the response to those needs is more often found in an institutional response grounded in complicated institutional characteristics.

As noted previously, this study focused on public postsecondary institutions within a single state, thus limiting the generalizability of the study. It is possible that Texas public postsecondary institutions might have responded differently to basic needs insecurities than institutions in other states. On the other hand, there have been innovative basic needs programs developed at numerous public postsecondary institutions in Texas, especially in community colleges. Some of these colleges have been listed in the Aspen top ten and won the Aspen award. As such, public postsecondary institutions within the state that are seeking the development or implementation of basic needs programs are encouraged to consult with these colleges for examples and types of basic needs programs. Nevertheless, future research should examine basic needs programs in other states and consider additional institutional characteristics to augment our understanding of institutional responses to students' unmet basic needs.

Secondly, this study did not gather data on the capacity of various basic needs programs. As a result, we are unable to assess if the programs operate at their full capacity to meet students' unmet basic needs or if the programs only reach some students, leaving others under- or unserved. Future research must examine the presence of programs and their capacity to deliver the services that students need.

Moreover, this study did not include the financial conditions of the institutions nor the funding sources of the basic needs program offerings. Future research should explore how colleges and universities mobilize and marshal their financial resources to fund various basic needs programs and services. As all colleges and universities have multiple and often

competing priorities, it would be important to examine the ways in which funding sources and priorities are associated with basic needs programs. Furthermore, future research should consider linking basic needs programs to institutional-level outcomes, such as student success measures (e.g., retention rate and graduate rate), to assess the effectiveness of basic needs programs. However, before these future research endeavors can be made, this study serves as our baseline effort to document and understand the multifaceted linkages between a variety of institutional characteristics and the likelihood of developing basic needs programs and services across Texas public postsecondary institutions.

Taken together, our results suggest that public postsecondary institutions could take on the challenges of implementing multiple basic needs programs to address students' unmet basic needs. Having an organizational structure, along with management capacity and prioritization, is imperative for an effective and context-appropriate institutional response to college students' basic needs [49].

**Author Contributions:** Conceptualization, L.K.Z., X.X. and T.S.; methodology, L.K.Z. and X.X.; software, X.X.; validation, X.X., J.M.H. and R.T.; formal analysis, X.X.; investigation, L.K.Z., X.X., J.M.H., R.T., N.E. and S.C.; resources, T.S.; data curation, L.K.Z. and J.M.H.; writing—original draft preparation, L.K.Z. and X.X.; writing—review and editing, L.K.Z., X.X., T.S., S.C., J.M.H., R.T. and N.E.; visualization, L.K.Z., X.X. and S.C.; supervision, T.S., L.K.Z. and X.X.; project administration, T.S. and X.X.; funding acquisition, T.S., L.K.Z. and X.X. All authors have read and agreed to the published version of the manuscript.

**Funding:** This work was supported by the ECMC Foundation [Grant #G-1906-12740 | College Success]. The APC was funded by the ECMC Foundation [Grant #G-1906-12740 | College Success].

**Institutional Review Board Statement:** We used secondary data, which does not require IRB approval.

**Informed Consent Statement:** Not applicable.

**Data Availability Statement:** The raw data supporting the conclusions of this article will be made available by the authors on request.

**Conflicts of Interest:** The authors declare no conflicts of interest.

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
