# Peer review of "Adaptive Complexity: Examining Texas Public Postsecondary Institutions’ Provision of Student Basic Needs Programs"

_2813-4346, doi:10.3390/higheredu3020015_

Round 1

Reviewer 1 Report

Comments and Suggestions for Authors

First of all, I think the topic is interesting and also some of the findings. However, I do think that your paper needs to be more elaborated to be published.

Why you choose to do a content analysis? How you conducted the content analysis? (since it is not particularly clear). 

In results section (page 7, lines 307-308), "and other student assistance programs (74.8%) are notable as well", due its relevance, I believe that you can identified and analyse these assistance programs. 

Even if the findings are interesting, I miss the relation to previous research. So, I suggest a better argumentation in relation to previous research.

Overall, I think these suggestions can improve even more the quality of the manuscript. 

Author Response

  1. First of all, I think the topic is interesting and also some of the findings. However, I do think that your paper needs to be more elaborated to be published.

Thank you for your review. We have addressed your comments in the document.

  1. Why you choose to do a content analysis? How you conducted the content analysis? (since it is not particularly clear).

The following has been added to explain why we choose to use content analysis of websites.

“A quantitative content analysis of institutional websites was conducted because of the important role that institutional websites have in representing the institution and they serve as a communication channel to students. The visual and textual information provided on websites is intentional and communicates institutional priorities (Carneiro & Johnson, 2014). The official websites are curated repository of information that students can use to find information about a variety of policies, programs, and services. Postsecondary institutions control their institutional websites and limit the information posted. Therefore, we would expect only those basic needs programs officially endorsed by the institution to be included on the website.” 

The following information has been added to explain how the content analysis was conducted.

“To ensure the accuracy, completeness, and reliability of the data collected, members of the research team were trained in the website data collection protocol. Data collection involved a series of systematic keyword searches utilizing the website search bar, often located on the landing page. The list of basic needs programs included as key words was based on previous research [33]. Additionally, the key words of “basic needs”, “student needs”, and “help” were also used. Starting at the landing page, the key word was entered. The results of the key words search were read and the links on the first page of results were followed to determine if the information provided was an indication of a basic needs program at that institution. Two members of the research team collected data from each institutional website independently and inter-rater reliability was closely monitored. When discrepancies in data collection were identified, a third member of the research team reviewed the website independently. When discrepancies remained, members of the research team met to review adherence to the data collection protocol and to come to a consensus on the data in question. This process continued until all inconsistencies were resolved.”

  1. In results section (page 7, lines 307-308), "and other student assistance programs (74.8%) are notable as well", due its relevance, I believe that you can identified and analyse these assistance programs.

Examples of the types of programs included under “other assistance” programs has been added.

“The category of “other assistance” referred to a variety of basic needs program offered at only a few institutions. Examples in this category include a help desk to direct students to off campus services, legal services for students, and clothing assistance.”

  1. Even if the findings are interesting, I miss the relation to previous research. So, I suggest a better argumentation in relation to previous research.

The authors have added to the discussion section by incorporating previous research into this section.

  1. Overall, I think these suggestions can improve even more the quality of the manuscript.

Thank you for your suggestions.

Reviewer 2 Report

Comments and Suggestions for Authors

The authors stated a complicated system model approach. What does this look like? Are there examples in the extant literature concerning your example? The Rouse book is cited, but then shouldn't costs and other computations be counted? Rouse spoke about an architecture; what is the architecture?

The complicated system model does not pose a research question. Were all models and tests designed before the study or is this an example of HARKing (hypothesizing after results are known)? More importantly, what do descriptive statistics not explain that most analysis, such as correlation, does not? If there is a complicated system model, then can one produce a model, with drivers and factors? Complicated systems design necessitates a qualitative understanding of the statistics; I would think finances would matter.

Needlessly complex statistics. Just because one can run a statistical test, does not mean one should. Not having the tables prevents a reviewer from assessing most of the results. Still, there are some problems.

1. Cronbach's alpha. The use is unheard of. Why is there reliability analysis here? This use is inappropriate and never seen in the extant literature.

2. Many statistical tests for what is really descriptive data. Many statistical tests greatly inflate the chance of Type 1 error. Giving descriptive statistics for most of the research would be far simpler and clearer.

3. Correlational analysis of the different programs seems problematic. Is there really a linear relationship that is practically significant (versus statistical significance)? For example, what is the practical relationship? Line 310, stating 98%, would be far simpler and more accurate. What value does the correlation add, and is there a meaningful relationship? [The 98% explains everything! Few food pantries do as well.]

4. Odds ratios of basic needs in different institutions, such as 2-year versus 4-year, are easily shown in the descriptive statistics. The extant literature could explain this fact as well, with residential vs non-residential largely influenced by type of institution. 

5. Comparing the same level of institutions with each other might be helpful. "Might" is a strong word because, with such high levels of all basic needs programs, there might be little variance. Explaining the difference by factors, which probably should include finances, could be much more helpful. I suspect, though, that most institutions are so similar that one would report that.

6. Correlation and/or logistic regression of factors that might have been hypothesized to have a linear relationship would still be applicable. One would have to hypothesize these beforehand. Correlation is not causation, but causation is correlation.

7. No tables visible means reviewers cannot assess any model, especially the logistic regression model (though one would be led to believe the descriptive statistics would show the same, especially with multicollinearity being a problem). Part of the question would be whether a more parsimonious model would explain the system as described. One would start with only size and residential. Everything else flows from the size, residential, and ultimately, the money. Prove this starting point wrong.

The paper seems needlessly complex. 

Author Response

Reviewer 2

  1. The authors stated a complicated system model approach. What does this look like? Are there examples in the extant literature concerning your example? The Rouse book is cited, but then shouldn't costs and other computations be counted? Rouse spoke about an architecture; what is the architecture?

The authors stated, consistent with Rouse, that “colleges and universities are complex adaptive systems consisting of both complicated system factors and complex system factors”. There are not examples that we are aware in the extant literature that use this element of Rouse’s theory to describe organizational response. Complicated adaptive systems are structural. The “architecture” is that of an interplay of ecosystem, structure, processes, and practices (Rouse, 2016, p. 2). We have revised the manuscript to clarify that structure influences processes and practices.

“Institutions exist within a broader societal and governmental ecosystem. The structure of the institution, consisting of the ways in which each institution is organized across academic and non-academic units, shapes policy and practices. This architecture of the system, as Rouse (2016) conceptualizes, and the elements comprising that architecture act to enable or inhibit other elements within the system. Economic elements are integrated into the architecture. We argue that has complex adaptive systems, comprised of complicated system factors and complex systems factors influence institutional responses in the forms of policies and practices. Further, complicated system factors and complex system factors function differently within a single institution. Given the nature of complicated system factors organizationally, it is more likely that those factors can produce formal organizational responses to situations such as unmet basic student needs.”

  1. The complicated system model does not pose a research question. Were all models and tests designed before the study or is this an example of HARKing (hypothesizing after results are known)? More importantly, what do descriptive statistics not explain that most analysis, such as correlation, does not? If there is a complicated system model, then can one produce a model, with drivers and factors? Complicated systems design necessitates a qualitative understanding of the statistics; I would think finances would matter.

The theoretical framing of the study occurred during conceptualization as we sought to understand differences in institutional responses based on organizational features. These are classified as complicated system factors and complex system factors. We have added information to help clarify the writing.

  1. Needlessly complex statistics. Just because one can run a statistical test, does not mean one should. Not having the tables prevents a reviewer from assessing most of the results. Still, there are some problems.

The tables were provided. It is not clear why they were inaccessible to the reviewer. We have attempted to address the broad issues identified by Reviewer #2 with particular attention to ensure that predictive language was removed.

  1. Cronbach's alpha. The use is unheard of. Why is there reliability analysis here? This use is inappropriate and never seen in the extant literature.

The mention of Cronbach’s alpha in the original document page 6 lines 263-264 has been removed.

  1. Many statistical tests for what is really descriptive data. Many statistical tests greatly inflate the chance of Type 1 error. Giving descriptive statistics for most of the research would be far simpler and clearer.

Thank you for this comment. On page 7 (lines 319-320), we added one sentence to address this potential problem.

“The statistical significance of p<.05 was employed to highlight the relative importance of the institutional characteristics.”

  1. Correlational analysis of the different programs seems problematic. Is there really a linear relationship that is practically significant (versus statistical significance)? For example, what is the practical relationship? Line 310, stating 98%, would be far simpler and more accurate. What value does the correlation add, and is there a meaningful relationship? [The 98% explains everything! Few food pantries do as well.]

One page 7 (lines 310-313), we stated:

“This analysis was conducted to help describe the tendencies of Texas public postsecondary institutions to offer multiple interrelated programs. As such, the underlying linear relationships among these variables were not emphasized.”

  1. Odds ratios of basic needs in different institutions, such as 2-year versus 4-year, are easily shown in the descriptive statistics. The extant literature could explain this fact as well, with residential vs non-residential largely influenced by type of institution.

We agreed with these observations. The purpose of using multivariate regression models is to estimate the independent effects of these characteristics after controlling for HSIs. On page 7 (lines 313-316), we stated:

For multivariate statistical analysis, binary logistic regression models were used to estimate the independent effects of the characteristics of Texas public postsecondary institutions on the likelihood of offering each basic needs program.

  1. Comparing the same level of institutions with each other might be helpful. "Might" is a strong word because, with such high levels of all basic needs programs, there might be little variance. Explaining the difference by factors, which probably should include finances, could be much more helpful. I suspect, though, that most institutions are so similar that one would report that.

The authors have included the reviewer’s suggestion into the limitations and future research.

“…this study did not include the financial conditions of the institutions nor funding sources of the basic needs program offerings. Future research should explore how colleges and universities mobilize and marshal their financial resources to fund various basic needs programs and services. As all colleges and universities have multiple and often competing priorities, it would be important to examine the ways in which the funding sources and priorities are associated with the basic needs programs.”  

Round 2

Reviewer 2 Report

Comments and Suggestions for Authors

Much improved, especially since I can see the tables. The major problem continues: The correlation is highly inappropriate and misleading. A first rule of correlation is to see if there is a linear relationship between two variables that suggests a causal mechanism. The relationships you present are spurious; while correlation is not causation, one infers that there is a relationship and some type of causal mechanism. 

Here are two examples: Ice cream and shark attacks correlate; diabetes and number of books in a house. No more can one reduce ice cream sales to reduce shark attacks than someone can sell more books to reduce diabetes, can one claim a meaningful relationship. Both can be graphed, but unlike other correlational designs, one cannot infer a meaningful relationship that getting a food panty correlates to childcare. Any basic statistical textbook will point these problems out.

A --> C is your design. B is postsecondary size, level of the institution, and residential. The correlations are spurious; we must move beyond p values for statistical significance. As stated last time, these variables are much more interesting and while obvious, could still be shown.

Parsimonious models are required. Correlation requires identifying what one perceives to be a causal relationship; there might be confounders and lurking variables unidentified, but the research is a starting point. Do the authors really believe there is a causal connection? If not, then what is the point.

Please consider the following direction: "If the dataset has two subgroups of individuals whose values for one or both variables differ from each other, this can lead to a false sense of relationship overall, even when none exists within each subgroup." Source below.

Aggarwal, R., & Ranganathan, P. (2016). Common pitfalls in statistical analysis: The use of correlation techniques. Perspectives in Clinical Research, 7(4), 187-190.

Either drop the correlation design or make a meaningful connection, such as size, level of institution, and residential OR other meaningful variable. Everything else should be fine. 

Here is another example of a proper correlation that started with a priori causal explanations and theory to predict meaningful relationships:

Nora, A. (1990). Campus-based aid programs as determinants of retention among Hispanic community college students. The Journal of Higher Education61(3), 312-331.

One still cannot resolve the cause-effect problem with correlation, but the purpose is to do the following: a.) Make a causal inference that b.) Could be further tested and, if possible a causal link established.

Thank you for dropping the inappropriate use of Cronbach's alpha and producing the tables. 

Author Response

Thank you for the additional review and guidance. We have removed the correlation including the original Table 2, first sentence in abstract, text in the Analytical Strategies subsection (originally lines 306-312) and text in the Results section (originally lines 344-354). 

Additionally, we have renumbered the original Table 3 to be labeled Table 2 and made relevant corrections in the text.

Round 3

Reviewer 2 Report

Comments and Suggestions for Authors

Much improved.